# Variation in the Root System Architecture of Peach × (Peach × Almond) Backcrosses

**DOI:** 10.3390/plants12091874

**Published:** 2023-05-03

**Authors:** Ricardo A. Lesmes-Vesga, Liliana M. Cano, Mark A. Ritenour, Ali Sarkhosh, Josè X. Chaparro, Lorenzo Rossi

**Affiliations:** 1Indian River Research and Education Center, Horticultural Sciences Department, Institute of Food and Agricultural Sciences, University of Florida, Fort Pierce, FL 34945, USA; ricardolesmes@ufl.edu (R.A.L.-V.);; 2Indian River Research and Education Center, Plant Pathology Department, Institute of Food and Agricultural Sciences, University of Florida, Fort Pierce, FL 34945, USA; lmcano@ufl.edu; 3Horticultural Sciences Department, Institute of Food and Agricultural Sciences, University of Florida, Gainesville, FL 32603, USA

**Keywords:** *Prunus*, stone fruit, rootstock breeding, stem cutting, root system architecture, rhizotron

## Abstract

The spatial arrangement and growth pattern of root systems, defined by the root system architecture (RSA), influences plant productivity and adaptation to soil environments, playing an important role in sustainable horticulture. Florida’s peach production area covers contrasting soil types, making it necessary to identify rootstocks that exhibit soil-type-specific advantageous root traits. In this sense, the wide genetic diversity of the *Prunus* genus allows the breeding of rootstock genotypes with contrasting root traits. The evaluation of root traits expressed in young seedlings and plantlets facilitates the early selection of desirable phenotypes in rootstock breeding. Plantlets from three peach × (peach × almond) backcross populations were vegetatively propagated and grown in rhizoboxes. These backcross populations were identified as BC1251, BC1256, and BC1260 and studied in a completely randomized design. Scanned images of the entire root systems of the plantlets were analyzed for total root length distribution by diameter classes, root dry weight by depth horizons, root morphological components, structural root parameters, and root spreading angles. The BC1260 progeny presented a shallower root system and lower root growth. Backcross BC1251 progeny exhibited a more vigorous and deeper root system at narrower root angles, potentially allowing it to explore and exploit water and nutrients in deep sandy entisols from the Florida central ridge.

## 1. Introduction

Given the continuing increase in food demand and the environmental impacts of agricultural production worldwide, the improvement of water and nutrient use efficiency is essential for sustainable horticulture. Plant genotype determines root morphology, which influences the plants’ efficiency in nutrients and water uptake [1]. There is widespread evidence for genotypic variation in root traits of many species [2] and the genus *Prunus*, to which peach [*Prunus persica* (L.) Batsch] belongs, is no exception. Peach is the third most produced temperate tree fruit species behind apple and pear [3].

Root architectural traits determine the temporal and spatial distribution of the root systems in soil, playing a fundamental role in the ability of plants to uptake soil resources. According to Manschadi et al. [2], root system architecture (RSA) is defined as the in situ space-filling properties or the spatial distribution of the root system within the rooting volume. Lynch [4] defines RSA as the spatial arrangement and growth pattern of roots, which influences the water and nutrient uptake ability of plants and their exploration capacity in the growing media in response to resource distribution. Thus, RSA determines the productivity and adaptation of horticultural crops to suboptimal soil environments [2,5]. Rootstocks with longer roots and numerous lateral branches and root hairs explore the soil more efficiently as they uptake water and nutrients at different depths and soil textures [6]. RSA traits play a fundamental role in achieving this goal, influencing the ability of root systems to take up water and nutrients [7]. This was confirmed by Fitter et al. [8], who demonstrated that nutrient and water uptake efficiency is a function of root system architecture. The development and configuration of the root system affect plants’ soil exploration and resource exploitation in the niche occupied by plants [9]. A herringbone root architecture (branches from a single primary root axis) is typically developed by plants adapted to nutrient- or water-poor soils [10].

In general, the main challenges for peach production are associated with drought, waterlogging, alkalinity tolerance, and soil-borne diseases, especially nematodes [11,12]. Since the *Prunus* genus encompasses a wide number of species (over 230), exploiting this genetic diversity can significantly enhance the development of rootstocks with improved resource use efficiency and field performance [13]. The range of rootstocks available for peach production has increased dramatically in the last few decades [3]. Despite the fundamental importance of studying root systems, this “hidden half” has not been studied as detailed as the aerial part of the plants, particularly in perennial fruit trees such as peach. Studying the RSA and diversity of root traits among peach genotypes can be important for rootstocks breeding given the horticultural applications for water and nutrient management of orchards [9,14]. The relative tolerance of rootstocks to water stress is influenced by RSA traits, such as rooting depth, root density, specific root length, and root/shoot ratio [15].

Extensive research has reported the potential for RSA improvement since traits such as root depth appear to be controlled by multiple genes in crops such as wheat [7]. Nevertheless, root growth and architecture are also influenced by the local soil environment, depending upon the plasticity of the genotype [16]. Soil is often heterogeneous and a complex medium with high spatial and temporal environmental variability (i.e., soil texture, structure, nutrient, and water content) [17,18]. There are contrasting RSAs between plant materials and soil types, as found in the peach production area in Florida. Root architectural traits that increase the acquisition efficiency for one soil resource may not be appropriate to capture other soils’ resources. Optimizing root architecture to improve the acquisition efficiency for one soil resource may incur trade-offs for acquiring other resources [2]. For instance, shallower RSAs are more efficient in acquiring immobile nutrients such as phosphorus (P). In contrast, deeper RSAs optimize water uptake and increase the acquisition of mobile nutrients such as nitrate (NO_3_-N) [5]. As pointed out by Manschadi et al. [2], the advantages of architectural root traits must be interpreted in the context of the type of environment in which the crops are grown. Therefore, it is highly relevant to identify the appropriate rootstock root system suitable to the specific limitations, and that can offer advantages that the in situ-soil type presents.

One of the particularities of the peach industry in Florida (USA) is that most of the main production areas cover contrasting soil types represented by psamments (sandy entisols) in the central ridge and aquods (wet spodosols) in the Flatwoods [19]. Since choosing the best rootstock for a particular soil is one of the most important decisions for successful peach orchard establishment [20], the study of rootstock RSA is crucial for understanding their adaptability to given edaphic conditions [21].

For the reasons given above, it is very important to develop studies that provide information about the potential horticultural performance of rootstocks in light of their RSA analysis. However, the study of the below-ground part of perennial fruit trees, such as stone fruits, under field conditions is highly challenging. The development of methods and equipment such as rhizoboxes has helped to overcome these issues by allowing direct and repeated observations of the roots within the rhizosphere [22,23]. Rhizoboxes also permit the imaging and monitoring of root growth dynamics without disturbance [1,24].

The total root length of a system is the most important measure of its growth [2,25,26,27]. According to Manschadi et al. [28], the analysis of early expressed root traits, such as root angle, represents potential selection criteria for breeding. Many studies of RSA analysis are carried out in young seedlings under laboratory conditions since seedling roots can be observed and measured rapidly and relatively easily. These laboratory conditions allow for higher detail, replication and standardization, allowing the comparison between species and genotypes [14].

‘Flordaguard’ is a seed propagated rootstock released by the University of Florida in 1991, with complex parentage that includes *P. persica* and *P. davidiana* in its pedigree. It is currently the only rootstock recommended for commercial peach production in Florida mainly because of its lower chill requirement and resistance to root-knot nematode, including *Meloidogyne floridensis* found in Florida soils [29]. Among other advantages that ‘Flordaguard’ exhibits, this rootstock is compatible with all peach cultivars, propagates easily by seed, has quicker readiness for grafting [30,31], and its red leaves facilitates the suckers detection and removal. However, ‘Flordaguard’ is susceptible to iron deficiency chlorosis under alkaline conditions [32]. The backcrossing with almond for peach production confers better adaptation to low-chill areas [33,34], and more tolerance to drought and alkalinity avoiding iron chlorosis (known as lime-induced chlorosis) [20,35,36]. In this study, ‘Flordaguard’ was used in one of the peach × (peach × almond) backcross families studied in this experiment. The peach × almond parental selections used in this study, being ‘Flordaguard’ the female parental line, are highly resistant to *Botryosphaeria dothidea*. These materials segregate rootstock populations for peach production on contrasting soils in Florida (USA): Sandy Entisols of the Central Ridge and Wet Spodosols of the Flatwoods [19]. The main objective of this study was to compare the root system architecture of vegetatively propagated peach × (peach × almond) backcrosses.

## 2. Results

### 2.1. Root Growth Parameters

The average total root length of Backcross BC1260 was 243 cm, which was significantly lower than that of BC1251 (806 cm) and BC1256 (591 cm) (Table 1). Similar differences were obtained for total root surface area between the backcrosses BC1251 (82.1 cm^2^), BC1256 (71.9 cm^2^), and BC1260 (29.4 cm^2^) (Table 1). There were no significant differences between the average root diameters (0.39–0.44 cm) of all the backcrosses (Table 1). The total root volume of BC1260 (0.29 cm^3^) was significantly lower than BC1251 (0.69 cm^3^) and BC1256 (0.72 cm^3^), which were not significantly different from each other (Table 1). BC1251 generated significantly more root tips (3608) compared with the backcrosses BC1256 (1920) and BC1260 (1156), which were also significantly different from each other (Table 1). These relative differences were also evident in the number of root forks, where BC1251 (3207) had a significantly higher number of root tips than backcrosses BC1256 (1803) and BC1260 (1179), which did not report a significantly different number of root forks from each other (Table 1).

### 2.2. Root Structural Parameters

The root specific length of BC1251 (6500 cm/g) was significantly higher than backcrosses BC1256 (5195 cm/g) and BC1260 (5534 cm/g), which show non-significant differences between each other. Similarly, the root fineness of BC1251 (1201 cm/cm^3^) was significantly higher than BC1256 (832 cm/cm^3^) and BC1260 (860 cm/cm^3^) (Table 1). Finally, the root tissue density of backcross BC1256 (0.16 g/cm^3^) was not significantly different from BC1251 and BC1260 (0.19 and 0.15 g/cm^3^, respectively). However, BC1251 was significantly higher than BC1260 for this structural root parameter (Table 1).

### 2.3. Root Dry Weight and Morphological Components

The root dry weight of BC1260 (0.04 g) was significantly lower than BC1251 (0.14 g) and BC1256 (0.12 g), which were not significantly different from each other (Table 1).

BC1251 had a root mass ratio (0.13 g/g) significantly higher than BC1260 (0.06 g/g) (Table 1). The root mass ratio of BC1256 (0.11 g/g) was intermediate and not significantly different from BC1251. These relative differences were mirrored in the root length ratio for BC1251 (661 cm/g), BC1256 (512 cm/g), and BC1260 (334 cm/g) (Table 1).

### 2.4. Total Root Length Distribution by Diameter Classes

The total length of very fine roots (≤0.5 mm) of BC1251 (673.41 cm) was significantly higher than BC1256 (442.04 cm), which was significantly higher than BC1260 (189.59 cm) (Figure 1a). However, the total length of fine (>0.5–≤1.0 mm) and thin (>1.0 mm) roots from BC1251 (118.59 cm and 13.60 cm, respectively) were not significantly different from BC1256 (129.84 cm and 19.23 cm). Conversely, BC1260 had significantly lower values in fine and thin class roots (47.29 cm and 6.62 cm) compared with the other two backcrosses.

### 2.5. Root Spreading Angle

The total root length of all the backcrosses were not significantly different within the shallower (0–25°) and shallow (25–45°) spreading angles (Figure 1b). However, the root length within the deep spreading angle (45–65°) in the backcross BC1260 (85.68 cm) was significantly lower than the other two backcrosses. Finally, within the deeper spreading angle (65–90°), BC1251 showed a significantly higher root length (352.10 cm) than the other two backcrosses, which were significantly different to each other, where BC1256 showed a higher root length (248.62 cm) than BC1260 (85.68 cm) (Figure 1b).

### 2.6. Root Depth Pattern and Root Depth Index

There were no significant differences between BC1251 (Figure 2a) and BC1256 (Figure 2b) within the percentage of total root length between their root horizons (A, B, and C). Conversely, in BC1260 (Figure 2c), the percentage of the total root length from horizon C (8.78%) was significantly lower than horizon A (51.49%) and horizon B (41.92%), which were not significantly different from each other. Regarding the root depth index (RDI), the backcross BC1260 (10.5) had significantly lower values than BC1251 (15.0) and BC1256 (14.4), which were not significantly different from each other (Figure 2d).

## 3. Discussion

Most of the root growth parameters of backcross BC1260 exhibited a smaller average size. A possible explanation for the smaller root system in BC1260 may be due to the vigor reduction provoked by inbreeding depression, where the reduction in vigor traits occurs in offspring of related parents [37,38]. BC1260 was the only backcross that has ‘Flordaguard’ in the pedigrees of the male (peach × almond male parent) and female parents (‘R95654.16’). The selection ‘R95654.16’ originated from a USDA selection × ‘Flordaguard’ F_2_ population. It is possible that backcross BC1260 is potentially a dwarfing rootstock. In peach, dwarf trees exhibit smaller root systems, and compact trees tend to exhibit high root branching [9].

On the other hand, the backcross BC1251 exhibited superior values in most root growth parameters resulting in a more robust root system in general. This is a potential advantage for BC1251 and BC1256 over BC1260 in the psamments (sandy entisols) of the Florida central ridge. The root system architecture is critically important for soil exploration and nutrient acquisition (Lynch, 2007), and trees with smaller root systems may be more sensitive to soil resource limitations typical of sandy entisols [9].

The backcrosses BC1251 and BC1256 exhibited an interesting superiority in root diameter classes. Fine roots (<1 mm in diameter) are believed to play an important role in water and nutrient uptake [39]. According to Solari et al. [40], the efficiency in water transport of peach rootstocks may be influenced by this trait, where most of the water uptake is presumed to occur in roots with <1 mm of diameter, with a direct effect on the radial hydraulic conductance. Moreover, BC1251 exhibited a higher number of root tips with a diameter <1 mm at the deepest horizon (C) compared with the other horizons and compared with the other backcrosses in this horizon (data not shown). This agreed with Basile et al. [41], who found in ‘K119-50’, a rootstock with almond in its genetic background, the highest number of fine roots in the deepest soil layer (below 69 cm) compared with ‘Nemaguard’ and ‘Hiawatha’ rootstocks.

Root dry matter and root morphological components of backcross BC1251 were higher in general, and significantly different from BC1260. This is especially advantageous for BC1251, since the root mass ratio can affect tree–soil–water relations [39]. Additionally, a higher root length ratio may suggest a more efficient soil exploration and higher hydraulic conductance, leading to superior water uptake efficiency [40]. Similar studies in citrus rootstocks have reported this correlation between a higher root length ratio and high root hydraulic conductance [42].

The root length per unit root biomass (root specific length) of BC1251 was also higher than the other backcrosses with no significant differences. Higher values of this trait suggest an efficient soil exploration at lower carbohydrate costs, which is beneficial under limited water supply, as reported by Eissenstat [43] in citrus rootstocks in sandy soil (quartzipsamment) in Florida. It is noteworthy that this soil type is similar to the psamments (sandy entisols) from the Florida central ridge, where a good part of Florida’s peach production is located. These soil types have a low water-holding capacity and essentially no horizon development or soil structure [19].

The root depth pattern of backcrosses BC1251 and BC1256 was similar, showing a deeper and more evenly distributed average root length between the substrate profiles. Conversely, the backcross BC1260 exhibited a significantly higher percentage of its total root length distributed in the shallowest horizon (A). This shallower root system would be more appropriate for flood-prone ‘Flatwood’ soils, where root asphyxiation can be a problem. Conversely, the root systems of BC1251 and BC1256 have a higher proportion of the roots in the deepest profile. This morphology suggests that BC1251 and BC1256 be evaluated in the deep, well-drained soils of the Florida central ridge. Such higher root-length distribution at depth allows potential access to water at greater depth, potentially making the rootstocks more drought-tolerant [28,44,45]. These features are also convenient in response to soil drying at the surface layers [2], as they tend to occur in psamments because of percolation [19]. Moreover, Glenn and Welker [46] found circumstantial evidence, indicating that the development of deep roots is important to maintain the root system when the soil’s top layers are dry in peach. In this sense, deeper root distribution patterns and higher root depth indexes are key for capturing leach-prone nutrients in such soils as well. A deeper and more vigorous root system enables access to leached nitrates (NO^3−^) and enhances drought adaptation by improving access to water stored in subsoil [45].

The higher values of root length distributed within the deeper spreading angles (65–90°) in the backcrosses BC1251 and BC1256 were consistent with the root distribution pattern and root depth index, confirming the feasible higher adaptability of BC1251 and BC1256 to deep sandy soils. According to Lynch [5], root gravitropism is an architectural trait under genetic control, and genotypes that express a narrow growth angle may be suitable for environments where plants rely largely on subsoil water [47]. Moreover, this type of root spreading may enhance the energy efficiency of these backcrosses for this purpose by inhibiting the elongation of lateral roots while maintaining primary root growth downwards [16,48].

Based on the principle of reiteration found in the entire root system of plum trees [49], we inferred the performance of adult rootstocks from the studied rootstock plantlets. Breeding for root system architecture traits may be a more efficient method to select for drought tolerance compared to breeding for physiological tolerance [36].

One of the goals of this study was to identify promising rootstock materials for the Florida central ridge and/or the Florida flatwoods. The root architecture of BC1251 and BC1256 is different to that observed in BC1260. Our results suggest that the three backcross populations should be evaluated under field conditions in locations with contrasting flatwood and central ridge-type soils. Such an experiment would demand the use of a technique different from rhizoboxes to study the RSA in situ with a wider time span. The results from such an experiment would indicate if the architectural data obtained in the rhizoboxes is predictive of root architecture in the field and, therefore, tree performance under varying soil conditions. Low-chill rootstocks that include plum (*Prunus* subg. *prunus*) in their genetic background are likely to be tolerant to flooding stress, whereas those crossed with almonds, such as the backcrosses BC1251, BC1256, and BC1260, are likely to be highly susceptible [50]. Finally, no anatomical studies were carried out on the rootstock materials used in this project. Therefore, we recommend for future studies to consider, in addition to the root architectural traits analyzed in this study, to include the additional factors such as the anatomical features of these backcrosses that may potentially influence the rootstock hydraulic conductivity in peach [40].

## 4. Materials and Methods

### 4.1. Plant Material

Seedlings of peach backcrosses: peach [*Prunus persica* (L.) Batsch] × (peach × almond [*Prunus dulcis* (Mill.) D.A. Webb]) were obtained. The mother trees were the peach selection ‘R95654.16’, located at the Southeastern Fruit and Tree Nut Research Laboratory of the USDA-ARS in Byron, GA, USA. The pollen parents were three different peach × almond hybrid selections: 1251, 1256, and 1260, located at the Fruit Tree Breeding and Genetics Laboratory of the Horticultural Sciences Department at the University of Florida in Gainesville, FL, USA. The parentages of the backcross families (BC) pedigrees are described in Table 2. The Peach × ‘Tardy Nonpareil’ almond hybrids used in this study have been identified as resistant to peach gummosis caused by *Botryosphaeria dothidea* [42].

### 4.2. Production of Backcross Populations

During the bloom season, “popcorn” stage (still closed and with expanded petals) were collected daily from peach × almond F_1_ hybrids for pollen extraction. The collected flowers were stored at 9 °C in sealed plastic bags, and the pollen was extracted by removing and drying the anthers at ambient temperature. The extracted pollen was stored in sealed plastic bags at 1 °C.

For controlled crosses, all open flowers were removed from the mother trees prior to pollination. Pollinations were performed daily on “popcorn” stage flowers. Flowers were emasculated by removing the sepals, petals, and stamens, exposing the pistil. Pollen was immediately applied to the stigma using a pencil eraser. Pollinations were performed over a 5-week period.

### 4.3. Seeds Stratification and Germination

The mature fruits were harvested from the mother trees, discarding the extracted seed that floated in the water. Non-floating seeds were hydrated by immersing them in water for 96 h, renewing the water every 24 h. After this period, the seeds were submerged in Captan fungicide (Drexel Chemical, Memphis, TN, USA) (0.15% *w*/*v*) for 24 h. Seeds were then stratified in a plastic bag containing perlite moistened with Captan (0.15% *w*/*v*) for six weeks at 4–8 °C, until germination. Germinating seeds were sown in plastic trays (720700C SureRoots^®^; T.O. Plastics, Inc.; Clearwater, MN, USA) containing a 1:1 blend of potting mix sphagnum (Jolly Gar-dener^®^ Pro-Line C/20 Growing Mix; Jolly Gardener Products, Inc.; Poland Spring, ME, USA) and coarse perlite (Specialty Vermiculite Corp.; Pompano Beach, FL, USA). Before filling up the germination trays, these were disinfected using a sodium hypochlorite solution (1.5% *v*/*v*) for 30 min. The potting blend was autoclaved at 121 °C for 90 min prior to use. The germination trays were covered with shade cloth at 70% shading in the greenhouse and watered manually. In the case of early fruit drops or pest/pathogen damage, the embryo rescue protocol developed by Chaparro and Sherman [43] was used: fruits were incubated in a sodium hypochlorite solution (1.88% *v*/*v*) for 20 min. Seeds were extracted under a laminar flow hood and cultured in test tubes containing sterile K_2_ tissue culture media with 30 mg of sucrose for peach ovule culture. Tubes were appropriately labeled, sealed with cellophane film (regenerated cellulose) and stored in a dark room at 4–8 °C until germination (radicle tip emergence).

### 4.4. Seedling Development

After germination, the seedlings were transplanted into plastic pots (10 cm × 10 cm × 35.5 cm) containing a 3:1 mixture of potting mix sphagnum and coarse perlite from the same manufacturers used for the germination trays. The plastic pots were previously disinfected using a sodium hypochlorite solution (1.5% *v*/*v*) for 30 min. The seedlings were grown within a plastic-covered greenhouse located in Fort Pierce, FL, USA, 27°25′34.2″ N–80°24′34.0″ W. Leaf color was used to rogue the seedling populations. Red leaved seedlings were hybrids and progeny from self-pollination green-leaved.

### 4.5. Softwood Cuttings Obtaining

Leafy softwood cuttings obtained from five selected 1-year-old plants of each backcross were treated with K-IBA at 0.2% (*w*/*v*) and rooted aeroponically, following the protocol described by Lesmes-Vesga et al. [51] (Table 2).

### 4.6. Plantlets Growing Conditions

After 28 days, three selected rooted cuttings (plantlets) from each BC plant were established in rhizoboxes (40 cm × 40 cm × 2.5 cm) to serve as biological replications. The rhizoboxes consisted of a three-sided plastic frame with two clear glass panes (40 cm × 40 cm) attached to both faces (front and back) (Figure 3a). The bottom of the rhizoboxes frames were perforated for drainage. The rhizoboxes were disinfected using a sodium hypochlorite solution (1.5% *v*/*v*) for 30 min prior to filling with a sphagnum/coarse perlite (3:1) potting mix (same manufacturers as germination trays substrate) containing 0.5% (*v*/*v*) Osmocote^®^ Plus (15-9-12) (A.M. Leonard, Inc., Piqua, OH, USA), a 3-month controlled release fertilizer. Additionally, a preventive control of pathogens was conducted prior to transplanting by drenching the substrate in each rhizobox and soaking the roots of the plantlets for 10 min in the systemic fungicide Luna^®^ Experience (Bayer CropScience, Research Triangle Park, NC, USA) [Fluopyram (17.6%) and Tebuconazole (17.6%)] diluted in distilled water at 0.2% (*v*/*v*).

The plantlets were transplanted in the center of the rhizoboxes to ensure their successful growth by avoiding air gaps around their base that generate root dehydration (Figure 3a). The glass panes of each rhizobox were covered with aluminum foil to keep the roots in the substrate under darkness. The rhizoboxes with plantlets were set standing in black plastic containers with bottom draining holes. The experiment was maintained under laboratory conditions (23 °C and 65% RH), where supplementary light from LED (light-emitting diode) bulbs and HPS (high-pressure sodium) lamps were installed to keep the photoperiod at 16–8 h (day-night). The plantlets were treated twice with a Luna^®^ Experience drench for pathogen control for the duration of the experiment. The plantlets were watered manually daily throughout the study.

### 4.7. Scanning of the Root Systems

The root systems of the plantlets were extracted from each rhizobox after 70 days for root analysis. The root system was accessed by removing one of the glass panes and inserting a pinboard into the roots-substrate mass to keep the spatial distribution of the roots. The pinboard consisted of a grid of acrylic panels with holes inserted on another panel with 2.5 cm-long nails separated 2.5 cm × 2.5 cm (Figure 3b). The pinboard-acrylic grid was inserted into the exposed side of the rhizoboxes, and the potting substrate was gently washed off the root systems inserted in the pinboard. The whole root system of each plantlet was scanned using a flatbed scanner EPSON Expression 10000XL (EPSON America, Inc.). Full-color images were captured in TIFF (Tagged Image Format File) at a resolution of 400 dpi (dots per inch). The aerial portion of each plantlet was removed, and the root systems were split into three horizons (A, B, and C) (Figure 3c). The roots from each horizon were placed in a Plexiglas tray (20 cm × 30 cm × 1.5 cm) containing water prior to scanning with an EPSON Perfection V800/V850 (EPSON America, Inc.) flatbed scanner. The Plexiglas tray with water was used to untangle the roots and minimize root overlap prior to scanning. The images were captured in TIFF at 600 dpi resolution.

### 4.8. Image Analysis Software and Measurements

The images obtained with the EPSON Perfection V800/V850 scanner were analyzed using the root image analysis software WinRHIZO^TM^ Pro (Regent Instruments Inc., Quebec City, Quebec, Canada). The root growth parameters measured were as follows: total root length (cm), total root surface area (cm^2^), average root diameter (cm), total root volume (cm^3^), number of root tips, and number of root forks. Additionally, total root length was distributed into three root diameter classes, following the criteria applied by Caruso et al. [52]: very fine (≤0.5 mm), fine (>0.5–≤1.0 mm), and large (>1.0 mm). The dry weight (g) of the aerial portion of the plant and the roots from each horizon was estimated by placing the samples in paper bags at 70 °C for 7 days, following the Ryser and Lambers [53] protocol.

Based on the root growth measurements and the dry weight of the aerial portion, the morphological components’ root mass ratio (RMR) (g/g) and root length ratio (RLR) (cm/g) were estimated. The RMR, which indicates the relative biomass allocated to the roots [53], was estimated by dividing the root’s dry weight by the whole plant’s dry weight. The RLR was estimated by dividing the total root length by the whole plant’s dry weight. This parameter expresses the root’s potential for the acquisition of below-ground resources. The structural root parameters were estimated as follows: root specific length (RSL) (cm/g), root fineness (RF) (cm/cm^3^), and root tissue density (RTD) (g/cm^3^). RSL was estimated by dividing the total root length by the root dry weight [54], the RF was estimated by dividing the total root length by the total root volume, and RTD by dividing the root dry weight by the total root volume.

The whole root system images, obtained with the flatbed scanner EPSON Expression 10000XL, were used to measure the root spreading angle (RSG) based on the protocol applied by Ramalingam et al. [55]. For this purpose, the root length (cm) was measured in four angular sections from the root systems images, being the cutting base the central point, with four categories as follows: shallower (0–25°), shallow (25–45°), deep (45–65°), and deeper (65–90°) (Figure 3d).

### 4.9. Experimental Design and Statistical Analysis

The experiment was arranged in a completely randomized design with five replications (*n* = 5) per backcross, using three plantlets per replicate. The collected data were analyzed by Analysis of Variance (ANOVA) using RStudio software (R, 2019), with a significance level of 0.05, and the means separation conducted using the Tukey HSD (Honestly Significance Difference) test.

## 5. Conclusions

The large genotypic diversity in *Prunus* is a valuable resource for breeding peach rootstock cultivars that overcome production challenges. Significant phenotypic differences in root system architecture of rootstocks that belong to the same genus may be found, with potential applications in their horticulture. This study demonstrates that similar populations can have different root architectures that may represent peach backcross. BC1251 represents a promising rootstock material, combining resistance to root-knot nematodes and peach gummosis (given its genetic background) with variable root architectures. As such, further field studies in these environments are warranted in addition to anatomical studies.

## Figures and Tables

**Figure 1 plants-12-01874-f001:**
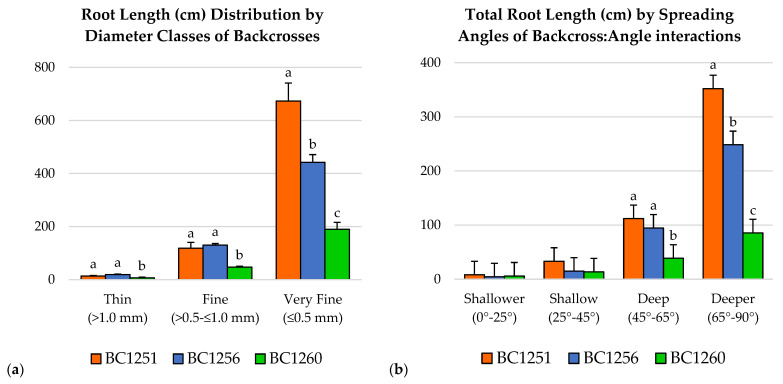
(**a**) Total root length comparisons grouped in three root diameter classes (very fine, fine, and thin) for families BC1251, BC1256, and BC1260, showing the interaction backcross/diameter class. (**b**) Root Spreading Angles (RSG) values estimated in terms of the total root length distributed between angular sections: shallower (0–25°), shallow (25–45°), deep (45–65°), and deeper (65–90°). Bars represent the standard error from the mean (*n* = 4), and different letters represent significant differences at *p* ≤ 0.05.

**Figure 2 plants-12-01874-f002:**
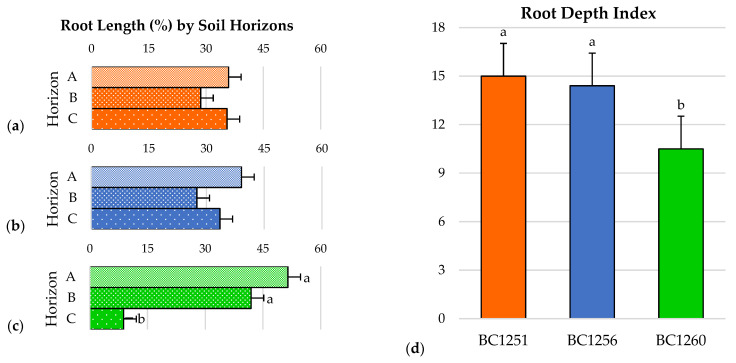
Root distribution pattern (RDP) based on the vertical distribution of roots of the backcrosses: (**a**) BC1251, (**b**) BC1256, and (**c**) BC1260 along the root system horizons (A, B, and C). (**d**) Root depth index (RDI) of the three backcrosses. Bars represent the standard error from the mean (*n* = 4), and different letters represent significant differences at *p* ≤ 0.05.

**Figure 3 plants-12-01874-f003:**
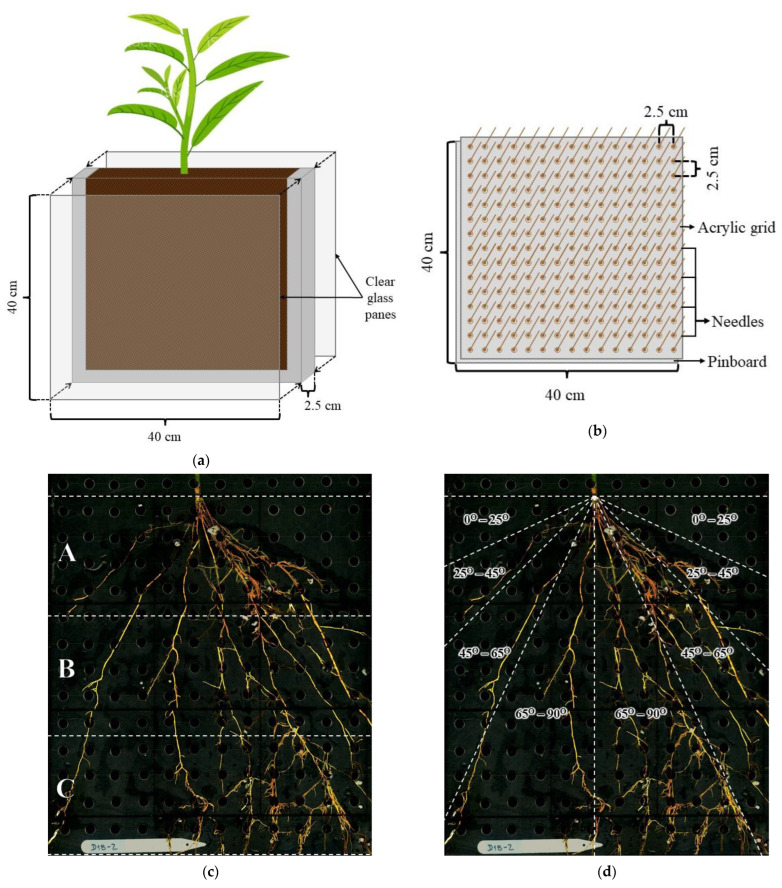
(**a**) Diagram of the rhizoboxes with their respective dimensions. (**b**) Diagram of the pinboard and the fitted acrylic grid to extract the root systems from the rhizoboxes. (**c**) Root system image of one of the root systems divided by horizons. The letters represent the three horizons (A, B, and C) in which the root systems were split for scanning. Each horizon had a depth of 10 cm. (**d**) Root system image of the root system, throughout the 3 horizons, divided by spreading angles to estimate the Root Spreading Angle (RSG). RSG was estimated by measuring the root length (cm) within each angular section: shallower (0–25°), shallow (25–45°), deep (45–65°), and deeper (65–90°). Photos courtesy of Ricardo A. Lesmes-Vesga.

**Table 1 plants-12-01874-t001:** Statistical results of the comparisons between the backcrosses BC1251, BC1256, and BC1260 for their root growth parameters, and structural parameters, dry weight, and morphological components. Mean values (*n* = 4) not connected by the same letter are significantly different according to Tukey’s studentized range (HSD) test (*p* ≤ 0.05).

Root Growth Parameters							
Backcross	Response	SE	Group	Backcross	Response	SE	Group
Total root length (cm)				Total Root Volume (cm^3^)			
BC1251	806	91.5	a	BC1251	0.69	0.10	a
BC1256	591	148.5	a	BC1256	0.72	0.16	a
BC1260	243	21.3	b	BC1260	0.29	0.02	b
Total Root Surface Area (cm^2^)				Number of Root Tips			
BC1251	82.1	10.6	a	BC1251	3608	41.85	a
BC1256	71.9	16.5	a	BC1256	1920	529.09	b
BC1260	29.4	0.5	b	BC1260	1156	164.60	c
Average Root Diameter (mm)				Number of Root Forks			
BC1251	0.41	0.06	ns	BC1251	3207	649.73	a
BC1256	0.44	0.04	ns	BC1256	1803	572.20	b
BC1260	0.39	0.03	ns	BC1260	1179	238.19	b
**Root Structural Parameters**				**Root Dry Weight and Morphological Components**			
Backcross	Response	SE	Group	Backcross	Response	SE	Group
Root Specific Length (cm/g)				Root Dry Weight (g)			
BC1251	6500	650.35	a	BC1251	0.14	0.03	a
BC1256	5195	377.91	b	BC1256	0.12	0.03	a
BC1260	5534	341.82	b	BC1260	0.04	0.00	b
Root Tissue Density (g/cm^3^)				Root Mass Ratio (g/g)			
BC1251	0.19	0.01	a	BC1251	0.13	0.03	a
BC1256	0.16	0.02	ab	BC1256	0.11	0.02	a
BC1260	0.15	0.02	b	BC1260	0.06	0.01	b
Root Fineness (cm/cm^3^)				Root Length Ratio (cm/g)			
BC1251	1201	47.15	a	BC1251	661	70.71	a
BC1256	832	133.90	b	BC1256	512	90.18	a
BC1260	860	123.63	b	BC1260	334	63.16	b

**Table 2 plants-12-01874-t002:** List of interspecific peach × (peach × almond) backcross populations from which the leafy softwood cuttings used in this study were obtained.

Backcross ID	Female ♀ (peach)	Male ♂ (peach × almond)
BC1251	‘R95654.16’	‘Fla. 97-47c’ × ‘Tardy-Nonpareil’
BC1256	‘R95654.16’	‘Fla. 97-42c’ × ‘Tardy-Nonpareil’
BC1260	‘R95654.16’	‘Flordaguard’ × ‘Tardy-Nonpareil’

## Data Availability

The datasets for this study can be provided upon request.

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
