# Peer review of "Variation in the Root System Architecture of Peach × (Peach × Almond) Backcrosses"

_plants, 2023, doi:10.3390/plants12091874_

Round 1
Reviewer 1 Report
Dear Authors,
I read with great interests your manuscript.
Firts of all, I think the description of the rhizoboxes is not clear, since it is not easy to understand.
Furthermore, I think the major concern is related to the use of a lot of table that can be resumed in few tables (or in graph, since with histograms I think it could be easier to understand the data).
Anyway, the manuscrit is quite clear and the results are relvevants. Also the experimental design sounds good.
Best regards
Author Response
The authors backcrossed peach cultivars, and investigated three of these backcrosses for differences in root morphology, with the goal to understand the plasticity present. This is important as adaptation to the different soil in Florida where peach is grown on. The study is carefully done, the methods well explained, the tables and figures well presented, and the text well written. Drawbacks in my eyes are the low number of biological replicates (4 plants per backcross), and the lack of a control line (like the parent lines) to compare the changes in root morphology to.
These limitations aside I think the data and methods presented are of interest to a broader readership. Detailed comments are listed below:
Thanks for your time and constructive comments. The paper has now been revised and our replies can be found below.
L46: overall, the introduction is well written and comprehensible. An aspect I miss a bit is in which areas peach root morphology should be improved most. What are the most pressing issues? Water uptake? N, P, K... starvation? this could be added in this paragraph.
This information is now added in that paragraph and in the beginning of the following.
L80: is there some information on how root morphology generally differs in these soils?
There is no information about how root morphology differs in these Psamments and Entisols in Florida.
L99: a bit more information could be added to this paragraph for readers not familiar with peach. why was it backcrossed with almond? and why is it the only rootstock recommended? some of this is mentioned in the methods section, but should be summarized shortly here. also, maybe explain the peach growing procedure shortly, like the grafting procedure, the age of the plants, when diseases occur...
This information is now added in the last paragraph of the introduction section.
L109: please explain shortly how the three backcrosses were identified. why did the study focus on these lines?
This information is now added in the last paragraph of the introduction section.
L197: some of this information would also be helpful in the introduction.
This information is now added in the last paragraph of the introduction section.
L263: please add some more information about the potential field experiment. would the trees grow longer in the field compared to the rhizotrons? is this expected to change the result? how do the backcrosses compare to the parent lines?
Based on the principle of reiteration mentioned at the beginning of the paragraph, the results are not expected to change.
L289: I would include some information of the methods section in the intro or results part, such as why the specific peach variety was chosen.
This information is now added in the last paragraph of the introduction section.
L306: reading the discussion, it would be interesting to know how the experimental soil used compares to filed conditions. e.g. is this a sandy soil?
No, the substrate used for germination was a blend of sphagnum:perlite (1:1).
L310: how many seeds were germinated for each backcross, and how often did the embryo rescue protocol have to be used?
There were 30 seeds germinated on average for each backcross, and the embryo rescue protocol was applied to less than 20% of them.
Results section: in the methods section, it is mentioned that the authors recorded the aboveground mass of the plants. I think this is not represented in a figure? could you include that? would be an interesting information.
The aboveground parts of the plants were not investigated in this study.
Tables: please add number of samples (n=...) to all tables and figures, where relevant.
The number of samples were added to the descriptive footnote of each corresponding table and figures.
Reviewer 2 Report
The authors backcrossed peach cultivars, and investigated three of these backcrosses for differences in root morphology, with the goal to understand the plasticity present. This is important as adaptation to the different soil in Florida where peach is grown on. The study is carefully done, the methods well explained, the tables and figures well presented, and the text well written. Drawbacks in my eyes are the low number of biological replicates (4 plants per backcross), and the lack of a control line (like the parent lines) to compare the changes in root morphology to. These limitations aside I think the data and methods presented are of interest to a broader readership. Detailed comments are listed below:
L46: overall, the introduction is well written and comprehensible. An aspect I miss a bit is in which areas peach root morphology should be improved most. What are the most pressing issues? Water uptake? N, P, K... starvation? this could be added in this paragraph.
L80: is there some information on how root morphology generally differs in these soils?
L99: a bit more information could be added to this paragraph for readers not familiar with peach. why was it backcrossed with almond? and why is it the only rootstock recommended? some of this is mentioned in the methods section, but should be summarized shortly here. also, maybe explain the peach growing procedure shortly, like the grafting procedure, the age of the plants, when diseases occur...
L109: please explain shortly how the three backcrosses were identified. why did the study focus on these lines?
L197: some of this information would also be helpful in the introduction.
L263: please add some more information about the potential field experiment. would the trees grow longer in the field compared to the rhizotrons? is this expected to change the result? how do the backcrosses compare to the parent lines?
L289: I would include some information of the methods section in the intro or results part, such as why the specific peach variety was chosen.
L306: reading the discussion, it would be interesting to know how the experimental soil used compares to filed conditions. e.g. is this a sandy soil?
L310: how many seeds were germinated for each backcross, and how often did the embryo rescue protocol have to be used?
Results section: in the methods section, it is mentioned that the authors recorded the aboveground mass of the plants. I think this is not represented in a figure? could you include that? would be an interesting information.
Tables: please add number of samples (n=...) to all tables and figures, where relevant.
Author Response
Dear Authors,
I read with great interests your manuscript.
Firts of all, I think the description of the rhizoboxes is not clear, since it is not easy to understand.
The rhizoboxes description was rephrased and the scheme of the box assembling was modified (Fig. 3a now).
Furthermore, I think the major concern is related to the use of a lot of table that can be resumed in few tables (or in graph, since with histograms I think it could be easier to understand the data).
Tables 1, 3 and 4 were merged in one single table (Now Table 1). The results from tables 2 and 5 were graphed and placed in Figure 1a and 1b now.
Anyway, the manuscrit is quite clear and the results are relvevants. Also the experimental design sounds good.
Thanks for your time.